# Comparing the effect of cross-group friendship on generalized trust to its effect on prejudice: The mediating role of threat perceptions and negative affect

Wahideh Achbari[1]*, Benny Geys[2,3], Bertjan Doosje[4]

1 Department of Political Science, University of Amsterdam, Amsterdam, The Netherlands, 2 Department of Applied Economics, Free University Brussels, Brussels, Belgium, 3 Department of Economics, BI Norwegian Business School, Oslo, Norway, 4 Department of Psychology, University of Amsterdam, Amsterdam, The Netherlands

* w.achbari@uva.nl

**Data Availability Statement:** Data cannot be shared publicly because of legal restrictions since the data are owned by a third-party organization. Data are available from the University of Bielefeld

## Abstract

Intergroup relations theory posits that cross-group friendship reduces threat perceptions and negative emotions about outgroups. This has been argued to mitigate the negative effects of ethnic diversity on generalized trust. Yet, direct tests of this friendship-trust relation, especially including perceptions of threat and negative affect as mediators, have remained rare at the individual level. In this article, we bridge this research gap using representative data from eight European countries (Group-Focused Enmity). We employ structural equation modelling (SEM) to model mediated paths of cross-group friendship on generalized trust via perceptions of threat and negative affect. We find that both the total effect as well as the (mediated) total indirect effect of cross-group friendship on generalized trust are weak when compared with similar paths estimated for prejudice.

## Introduction

Immigration has recently (once more) become a hot topic, and many empirical studies on the social implications of rising ethnic diversity in Western societies have appeared across the social sciences. Putnam's study was one of the first to suggest a detrimental effect of ethnic diversity in neighbourhoods on generalized trust [1]. At least 77 articles examining this link at different geographical levels have appeared since then [2, 3]. Much of this literature argues that the negative effect of diversity is linked to feelings of threat regarding out-groups, and some see intergroup contact as a possible remedy. The bulk of these studies takes the proximity of ethnic groups as a proxy for intergroup contact, even though such proximity has also been linked to intergroup conflict and to a constrict mechanism (whereby diverse contexts turn people into turtles shying away from public life altogether). Moreover, neighbourhood diversity indices do not reveal the dynamics of interethnic contact, which lie at the heart of intergroup relations theory [4].

(contact via Prof. Dr. Andreas Zick (zick@uni-bielefeld.de) or Prof. Dr. Beate Küpper (beate.kuepper@hs-niederrhein.de)) for researchers who meet the criteria for access. The dataset in our manuscript is titled: "Group-Focused Enmity in Europe", which is managed by Prof. Dr. Andreas Zick and Prof. Dr. Beate Küpper. More information about this dataset can be found at https://www.uni-bielefeld.de/(de)/ikg/projekte/GFE.html. The authors had no special access privileges that others would not have.

**Funding:** The authors gratefully acknowledge funding by The European Commission's H2020 Marie Skłodowska-Curie Actions Individual Fellowship (WA: grant number 750832; https://ec.europa.eu/research/mariecurieactions/actions/individual-fellowships_en) and Research Foundation – Flanders FWO (BG: grant number G.0022.12; https://www.fwo.be/). The funders had no role in study design, data collection and analysis, decision to publish, or preparation of the manuscript.

**Competing interests:** The authors have declared that no competing interests exist.

A few exceptions notwithstanding [5–10], cross-group friendship ties have not widely been modelled in the analysis of generalized trust. Yet, cross-group friendship meets key requirements specified by Allport [11] as crucial conditions for the contact hypothesis to work: i.e. friendship mitigates (perceived) social status differentials between the involved individuals, friends work together to achieve common goals, and despite societal segregation, friendship along ethnic lines is not prohibited in the societies we examine. While social psychological research has focused on the mediated paths between cross-group friendship via perceptions of conflict and out-group trust [12, 13], mediated paths to generalized trust have not been examined. We thus believe that a major limitation of prior research is the lack of a mediation analysis of contact (specified as cross-group friendship), perceived conflict, and generalized trust at the individual level.

Two recent meta-analyses and a large social-psychological literature on the relation between intergroup contact and prejudice further underscore our individual-level approach and the need for a mediation analysis. First, one forthcoming meta-analysis "indicates that ethnic diversity in residential settings does not lead to 'contact effects' under general circumstances" [2 p. 24.10], while another rejects both a "blanket version of the contact hypothesis (. . .) [and] a universal threat argument". Recent meta-analyses also convincingly maintain that contact and conflict mechanisms are complementary rather than incompatible at the individual level [2, 3, 14]. Second, several social-psychological studies on the relation between intergroup contact and prejudice [12, 15] as well as out-group trust [13] highlight the importance of mediating variables such as perceptions of threat and negative affect. Taking inspiration from this work, we argue that any effect of generalized trust on cross-group friendship may likewise run via lower levels of perceived conflict (threat and negative affect) at the individual level. There are important conceptual differences between generalized trust (trust towards people we do not know) [16], particularized trust (trust towards people we know) [17], and out-group trust. The latter is trust in ethnic groups one believes not to share a common descent with [2]. Generalized trust is the belief that unknown people do not do us harm [18], which differs from prejudice. Prejudice is "an antipathy based upon a faulty and inflexible generalization"[11 p. 9]. It involves the rejection of and opposition to contact with the out-group. Pettigrew and Meertens [19 p. 58] add to this definition that "prejudiced attitudes tend to form ideological clusters of beliefs that justify discrimination".

In theory, once durable intergroup contact such as cross-group friendship is in place, it should reflect in a strong negative relation to feelings of threat and negative affect. If a strong positive association with generalized trust is then paramount, we can directly assess whether the interplay between cross-group friendship and perceived conflict stipulated by intergroup relations theory is at work. These relations reflect central assumptions underlying prior research on ethnic diversity in different geographical locations. Alternatively, generalized trust may be less affected by cross-group friendship and perceived conflict than generally assumed. If so, (lack of) intergroup contact may not be the mechanism behind the creation and loss of generalized trust, and the ecological validity of previous work in this direction might be limited. To our knowledge no study has simultaneously tested the link between cross-group friendship, feelings of threat and negative affect, in relation to generalized trust across individuals in multiple countries. Needless to say, establishing such links is crucial for informing future public policy as well as advancing the nexus between generalized trust and intergroup relations theory invoked by social science scholarship.

Our empirical analysis of the relationship between cross-group friendship and generalized trust relies on representative survey data from eight European countries [20]. We explicitly include feelings of threat and negative affect as mediators. Whereas threat perceptions are embedded in a fear that the out-group undermines the very existence of the in-group in

economic, political, and cultural terms [21], negative affect is "a general dimension of subjective distress (. . .) that subsumes a variety of aversive mood states, including anger, contempt, disgust, guilt, fear, and nervousness" [22 p. 1063]. In a first step, we assess to what extent generalized trust, threat perceptions, negative affect, and prejudice are separate latent constructs as well as how strongly these constructs are related. This is done using a Confirmatory Factor Analysis (CFA) framework, and is essential for the evaluation of the measurement part of our structural models that follow.

Then we examine how cross-group friendship is related to generalized trust, both directly as well as indirectly via perceptions of threat and negative affect. While our main contribution lies in estimating paths between cross-group friendship and generalized trust, we also include prejudice in the model–operationalised as respondent's desired social distance from out-group members [23, 24]–as a second dependent variable. This allows us to cross-validate our findings with existing social-psychological studies, and compare the friendship-trust paths to those between friendship and prejudice to derive more detailed inferences on the observed effect sizes (see below). Our main findings indicate that the relation between cross-group friendship and trust turns out to be weaker than the relation between cross-group friendship and prejudice established in foregoing research.

The remainder of the article is organized as follows. We first synthesize theoretical arguments based on existing work and derive our hypotheses. Then we describe our data and the operationalization of our central variables, before turning to our key findings. Finally, we conclude the paper by discussing the implications of our results and suggesting avenues for further research.

## An integrated intergroup relations theory

**The role of cross-group friendship.** A substantial body of social-psychological literature indicates that intergroup contact is important for improving social relations between ethnic groups, which has given rise to an integrated intergroup relations theory [21, 25]. Intergroup contact studies show that it generally reduces prejudice [26–28], while segregation feeds threat perceptions and the persistence of negative emotions. Moreover, according to contact theorists, friendship across ethnic groups or a longstanding close relationship should mitigate perceived conflict [15, 27–29 p. 259]. However, a key question is whether such effects spill over to generalized trust [2, 15]. Generalized trust not only requires reduced prejudice or lower levels of negative emotions towards a specific out-group. It also implies extending this towards unknown people. Some argue that intergroup theory is applicable to the analysis of generalized trust due to 'secondary transfer effects'. These occur when positive attitudes towards encountered individuals are generalized to a wider group [15, 30]. If so, cross-group friendship may enhance a broader willingness to trust [31]:

H1$_A$: A strong relationship exists between cross-group friendship and generalized trust (total effect) comparable in size to that between cross-group friendship and prejudice.

Existing studies on this friendship-trust relation, however, provide conflicting conclusions. Phan [10] examines this link in Canadian neighbourhoods and cities, and finds no support for the contact theory. Another Canadian study [8] finds support for the contact theory only for younger cohorts. A possible explanation offered by the authors is that younger generations have grown up in a more multiculturally diverse environment compared to older generations, which might amplify the role of intergroup contact among younger generations. In Germany, Stolle et al. [7] surprisingly find that only weaker intergroup ties (such as conversation) matter, rather than strong friendship ties. Finally, Uslaner [6], Dinesen and Sønderskov [9], and Van der Linden, Hooghe, De Vroome, and Van Laar [5] again find at best mixed evidence. Since

these existing studies on the friendship-trust relation provide conflicting conclusions, our alternative hypothesis is:

H1$_B$: The relationship between cross-group friendship and generalized trust (total effect) is weaker than that between cross-group friendship and prejudice.

**Threat perceptions and negative affect as mediators.**   Importantly, research has yet to include mediators in the relationship between cross-group friendship and generalized trust at the individual level. As argued before intergroup contact reduces perceptions of threat towards–and negative emotions about–individuals of different ethnic backgrounds [1 p. 149, 15]. Actual or perceived competition over economic and symbolic resources (such as jobs, territory or values) and access to public goods may lead to feelings of threat among ethnic groups [21, 32–35]. Competition may thus trigger perceptions of threat, which can hamper intergroup relations [32] and increase prejudice [36]. However, social-psychologists have repeatedly shown that perceptions of threat and negative emotions mediate the effect of cross-group friendship on *prejudice* [15, 27–29]. The reason is that both elements are likely to impact upon the feelings of insecurity that accompany interactions with out-group members [28]. Schmid, Al Ramiah, and Hewstone [13] confirm that cross-group friendship reduces threat perceptions and fosters out-group trust. This mediating role of threat perceptions and negative affect may well carry over to the relation between cross-group friendship and generalized trust. Indeed, Bäck, Söderlund, Sipinen, and Kestilä-Kekkonen [37] argue that people high in generalized trust are likely to believe that immigrants are an asset to the welfare, culture or economic prosperity of a society, because generalized trust is associated with an optimistic worldview and a relative sense of control over one's environment [38]. Inattention to these mediators is particularly puzzling since many neighbourhood studies explicitly draw on the conflict hypothesis, but not always have specifically modelled its underlying mechanisms in their analyses of generalized trust [2, 3, 14]. We expect:

H2$_A$: The effect of cross-group friendship on generalized trust is mediated by negative affect and threat (total indirect effect), which is comparable in size to that of cross-group friendship on prejudice.

In contrast, Dinesen, Schaeffer, and Sønderskov [2] argue that since generalized trust is based on the "evaluations of aggregates of people without a specific ethnic group component" (i.e. most people), a conflict mechanism may be less straightforwardly applicable. We found only one study that included cross-group friendship as well as threat perceptions as mediators in the analysis of generalized trust. This study based on results across Australian local communities reports a weak path [39]. Our alternative hypothesis is therefore as follows:

H2$_B$: The effect of cross-group friendship on generalized trust is mediated by negative affect and threat (total indirect effect), which is weaker than that of cross-group friendship on prejudice.

**Direction of causality.**   A few words on the direction of causality are required. Brown and Hewstone's [29] review of the literature lists three reasons why contact is an antecedent of prejudice, and why causality does not run in the other direction–at least initially. First, several empirical studies using non-recursive Structural Equation Modelling find supportive evidence for this order. Second, experimental studies in which participants have no option for choosing an out-group member tend to display larger effect sizes than studies where participants do have such a choice. In such settings, observed effects cannot be the result of the avoidance of contact by prejudiced people [28], which suggests that causality runs from contact to prejudice. Third, most longitudinal studies confirm this direction of causality [23, 40].

There are also a few longitudinal studies testing whether perceptions of threat are antecedents of contact, or whether prior contact may shape these attitudes. Brown and Hewstone [29] summarize these studies and conclude that cross-group friendship ties are predictors of

reduced threat perceptions [23, 41–43]. Swart, Hewstone, Christ, and Voci [44] furthermore show that over time cross-group friendship ties lead to less anxiety about the out-group, which decreases prejudice. As far as threat perceptions are concerned, Schlueter and Scheepers [45] show these are causal antecedents of Germans' dislike and negative behavioural intentions towards foreigners as well as Russians' ethnic distance towards minorities. Finally, in a network study of friendship and prejudice, Stark [46] shows that prejudiced students are not actively avoiding cross-group friendship, which confirms the experimental results discussed above. Although prejudiced people may have less opportunity for contact and out-group friendship, once contact interventions are in place these reduce prejudice outside the laboratory [47].

That being said, our cross-sectional analysis cannot establish causality. Although one might pose that investigating the relations under analysis in a cross-sectional survey is futile, we believe our analysis nonetheless retains significant merit by assessing and highlighting the potential role of mediators in the contact-trust relation at the individual level.

## Methods

### The dataset

Our dataset is derived from the Group-Focused Enmity study [20], which was conducted in eight European countries in winter 2008/09 by TNS Infratest and their European partners. This dataset is ideal for our purpose, since it allows for a multiple-country analysis with consistent measures of cross-group friendship and generalized trust, as well as the set of theoretically important mediators (i.e. threat and negative affect). The analysis includes data from Britain, France, Germany, the Netherlands, Italy, Portugal, Poland, and Hungary. In each country, a representative sample of 1,000 respondents– 16 years and older–was interviewed via a telephone survey. Our analyses, however, only include the 85% of respondents without a migration background (neither themselves nor their parents or grandparents). This focus is driven by the fact that the effect of positive intergroup contact primarily occurs among the majority population [26, 48 pp. 954–5].

### Dependent and independent variables, and mediators

Table 1 describes the question wording of items that comprise our variables of interest, the scales, and their distribution (proportions, means, and standard deviations). Generalized trust–our first and central dependent variable–refers to a norm, which reflects expectations about the trustworthiness of unknown people or strangers [38, 49]. We employ the two items 'Trust' and 'Abuse' originally introduced by Rosenberg [50]. Previously, there has been some debate whether these items tap into trustworthiness of unknown people or measure trust towards familiar persons. Delhey, Newton, and Welzel [16] demonstrate that in most western countries the radius of "most people" in the trust question indeed includes people one has met for the first time, and people of different religions and nationalities.

To cross-validate our findings on generalized trust with the existing social-psychological literature on intergroup contact and prejudice, we include a second dependent variable reflecting prejudice. Since prejudiced attitudes are those that indicate social distance [23, 24], we use a scale, which was adapted from Pettigrew and Meertens [19]. These questions are indicative of increasing levels of social distance rejecting immigrants in the school of one's child, as a neighbour, as a colleague and to enter one's country. All these items are measured on 4-point ordinal scales that range from mostly disagree to mostly agree.

In the operationalization of our two mediators, we rely on scales adapted from Stephan et al. [51] for threat perceptions, and Watson, Clark, and Tellegen [22] for negative affect. The

**Table 1. Variables of interest and their descriptive statistics.**

| Variables | Question wording | Distribution |
|---|---|---|
| **Dependent** | | |
| **Generalized Trust** | | |
| Trust | Can most people be trusted, or would you be careful? | 63%: Careful; 37%: Trusted |
| Abuse (Reversed) | Do most people try to take advantage of you, or do they behave decently? | 31%: Advantage; 69%: Decently |
| **Prejudice** | (Unbalanced scales, 4 points from 'totally disagree' to 'totally agree') | |
| Social Distance (School) | Reluctant to send children to a school with a majority of immigrants | Mean = 2.6<br>St. Dev. = 0.99 |
| Social Distance (District) | Reluctant to move into a district with many immigrants | Mean = 2.6<br>St. Dev. = 0.99 |
| Social Distance (Work) | An employer should have the right to only employ native [country x's nationals] people | Mean = 3.2<br>St. Dev. = 0.90 |
| Social Distance (Country) | Vote only for parties that want to reduce the influx of immigrants | Mean = 2.9<br>St. Dev. = 0.99 |
| **Mediators** | | |
| **Threat** | (Unbalanced scales, 4 points from 'totally disagree' to 'totally agree') | |
| Economy | Immigrants who live here threaten the [country x's national] economy | Mean = 3.0<br>St. Dev. = 0.88 |
| Finance | Immigrants who live here threaten my personal financial position | Mean = 3.3<br>St. Dev. = 0.82 |
| National Values | Immigrants who live here threaten our way of life and our values [in country x] | Mean = 3.3<br>St. Dev. = 0.82 |
| Personal Values | Immigrants who live here threaten my personal way of life and my values | Mean = 3.3<br>St. Dev. = 0.80 |
| **Negative affect** | (Unbalanced scales, 4 points from 'not at all' to 'extremely') | |
| Fear | When you meet or think about immigrants, what do you feel? | |
| | Fear? | Mean = 3.6<br>St. Dev. = 0.71 |
| Anger | Anger? | Mean = 3.7<br>St. Dev. = 0.59 |
| Disgust | Disgust? | Mean = 3.8<br>St. Dev. = 0.52 |
| Relaxed (Reversed) | Relaxed? | Mean = 3.1<br>St. Dev. = 0.86 |
| **Main independent variable** | | |
| **Friendship** | How many of your friends are immigrant? | Mean = 1.9<br>St. Dev. = 0.89 |

Note: When the word native is stated above, in the questionnaire it read as the respective country's nationals (e.g., French, Dutch, etc.). Similarly, country x read as the respective country's name (e.g., France, The Netherlands).

latter "subsumes a variety of aversive mood states, including anger, contempt, disgust, guilt, fear, and nervousness" [22 p. 1063], while perceptions of threat are related to the decline of an entire group's economic, political, and symbolic power. Finally, since cross-group friendship or a long-term close relationship appears critical to the de- and recategorization processes

underlying the contact-prejudice relation [27–29 p. 259, 42], contact is operationalized as the respondent's frequency of having immigrant friends. The specific question originates from Brown [52].

## Control variables

We have included six control variables in the structural part of the analysis below. The first is gender, with 47% male and 53% women. The second is age (in years; M = 48.8, SD = 17.02) and the third is educational attainment, which is regrouped into three categories: *(i)* primary school or low vocational degree (24%); *(ii)* secondary school or vocational degree (44%); *(iii)* high vocational or university degree (28%). For simplicity, the latter is included as an ordinal scale. Fourth, we include household income (measured in ten income bands). The fifth variable is marital status, which we have regrouped into three categories: *(i)* married (56%), *(ii)* single (19%), and *(iii)* separated, divorced, or widowed (25%). Finally, we include dummies to capture any effects that arise from country differences (e.g., differences in the aggregate proportions of immigrants in each country).

## Modelling approach

Before testing our hypotheses, we first assess the relationship between generalized trust, prejudice, negative affect, and threat (i.e. the measurement model) in a Confirmatory Factor Analysis framework. Each set of items should form a separate latent construct and the overall model should have a good fit. Next, we test our hypotheses by evaluating direct, indirect, and total (indirect) effect in a set of Structural Equation Models. Strictly speaking, we cannot speak of "effects" since we employ cross-sectional data and the data available to us do not allow a credible resolution to the endogeneity concerns arising in our setting. As such, the analysis below cannot confirm the direction of causality. While we use Structural Equation Modelling terminology to avoid confusion, it should be kept in mind that our analyses merely establish the presence (or absence) of correlations between the variables under analysis. In our models cross-group friendship is included as an exogenous or independent variable with generalized trust and prejudice as dependent variables. Then the mediators threat and negative affect are added. We elaborate on the rationale behind these models in the results section, and here briefly pinpoint other general modelling considerations.

From a methodological perspective, it is important to note that most continuous variables are normally distributed. Overall, we use the default option to estimate the models using all available data. Hence, for the measurement model with the latent constructs, we have not deleted the missing values listwise (note that the proportion of missing values per item ranges from 0.5% to 8%). Our structural models, however, do not include cases with missing values for the negative affect factor. This has happened listwise and excludes 1,908 cases at random from the analysis (leaving 5,333 cases). This choice was driven by the questionnaire design; the negative affect questions were only asked from a random subset of respondents.

We estimated the models using *Mplus* 8.4. Since our models contain categorical data, we used matrices of polychoric correlations. The estimator of models with categorical data is Weighted Least Squares Means and Variance adjusted (WLSMV). For each latent factor, the loading of the first item is taken to be one and the error terms of the items are also taken to be one unless modified. To evaluate the overall goodness of fit of each model, we included several indices, and the following are cut-off points for good models: Chi-square not significant; *RMSEA*≤0.05 (≤0.08 for moderate fit); *CFI*≥0.95 (≥0.90 for reasonable fit); *SRMR*≤0.08 [53–58]. All reported effect sizes are *β*-coefficients (standardized) and therefore comparable. Unless otherwise stated, the discussed coefficients are statistically significant with p-value<0.001. In

addition, we report Bias corrected Bootstrapped Confidence Intervals (2000 iterations) for our total, total indirect, and indirect effects in order to compare the effect sizes. The Bootstrapped Confidence Intervals (BC CI) outperforms other tests since it does not assume that the data are multivariate normal, is not sensitive to sample size, and it is not symmetric around the estimated mediating effect. The latter is important because even if two effects are statistically significant their joint test is not necessarily so [59 p. 231, 60 p. 95, 61 p. 1].

## Results

### Measurement model

To determine the reliability and validity of our measurement model we compare the interrelationships between generalized trust, prejudice, threat, and negative affect across two models. Model 1 takes all the items to be part of one latent factor. Model 2 (Fig 1) separates the four factors, which are also taken to be correlated. Modification indices show improved model fit in all models when including correlated errors between Social Distance at District and Social Distance at School. The inclusion of this correlation is methodologically justified since these items follow each other in the questionnaire, which may have induced order effects.

The logic behind assessing these two models is twofold. First, they allow testing whether there is one simple latent construct rather than several latent constructs behind these items. For example, while model 1 assumes one latent factor, model 2 treats generalized trust, prejudice, threat and negative affect as four latent factors. Testing these models thus allows assessing how well the items group together and how they relate to the other constructs in the analysis. Secondly, identifying how well these items group together is important to accurately gauge

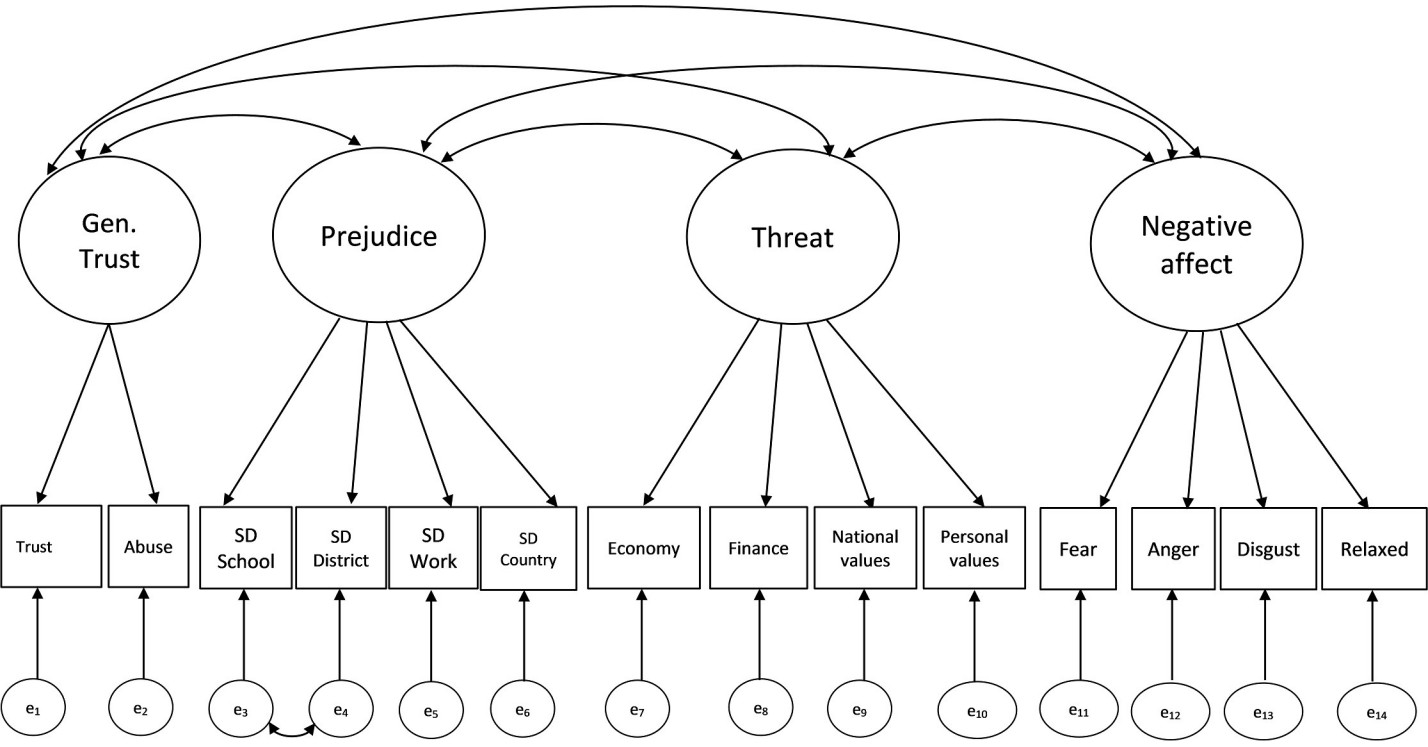

**Fig 1. Graphical representation of the measurement model.** Note: Model 2 in Table 2.

their relation to cross-group friendship. Otherwise, misspecifications of the structural equation models in the next section could be due to poor fit of the factors [62].

The results in Table 2 suggest that model 2 –with four separate factors–is empirically superior. It outperforms model 1 on all fit indices. Note that the factor loadings and the corresponding $R^2$ are worse in the first model, too. Overall, these results suggest that generalized trust, prejudice, threat and negative affect are separate latent factors, which correlate moderately. Their reliability as indicated by Cronbach's alpha is high (>0.7). Albeit acceptable, the generalized trust items' reliability is lower (0.5), because the dataset contained only two items.

**Table 2. The measurement model.**

| | Model 1 | | Model 2 | | |
|---|---|---|---|---|---|
| | $\beta$[a] | $R^2$ | $\beta$[a] | $R^2$ | Cronbach's alpha |
| **Items:** | | | **Gen. Trust** | | 0.503 |
| Trust | 0.392 | 0.154 | 0.870 | 0.757 | |
| Abuse (Reversed) | 0.315 | 0.099 | 0.659 | 0.435 | |
| | | | **Prejudice** | | 0.734 |
| SD[b]-School | -0.433 | 0.188 | 0.464 | 0.215 | |
| SD[b]-District | -0.472 | 0.223 | 0.508 | 0.258 | |
| SD[b]-Work | -0.658 | 0.433 | 0.752 | 0.566 | |
| SD[b]-Country | -0.558 | 0.312 | 0.625 | 0.390 | |
| | | | **Threat** | | 0.833 |
| Economy | -0.697 | 0.486 | 0.749 | 0.561 | |
| Finance | -0.663 | 0.440 | 0.724 | 0.524 | |
| National Values | -0.710 | 0.505 | 0.766 | 0.586 | |
| Personal Values | -0.669 | 0.448 | 0.727 | 0.528 | |
| | | | **Negative affect** | | 0.691 |
| Fear | -0.451 | 0.204 | 0.586 | 0.343 | |
| Anger | -0.547 | 0.299 | 0.727 | 0.529 | |
| Disgust | -0.520 | 0.270 | 0.681 | 0.464 | |
| Relaxed (Reversed) | -0.430 | 0.185 | 0.519 | 0.269 | |
| | | | **Correlation coefficient** | | |
| Prejudice ↔ Gen. Trust | | | -0.296 | | |
| Prejudice ↔ Threat | | | 0.783 | | |
| Gen. Trust ↔ Threat | | | -0.416 | | |
| Threat ↔ Negative affect | | | 0.606 | | |
| Negative affect ↔ Gen. Trust | | | -0.318 | | |
| Negative affect ↔ Prejudice | | | 0.646 | | |
| SD[b]-School ↔ SD[b]-District | | | 0.477 | | |
| **Model fit indices:** | | | | | |
| $\chi 2$ | 4965.254 | | 1119.445 | | |
| df (p) | 77 (0.000) | | 70 (0.000) | | |
| RMSEA (CI) | 0.094 (0.091 0.096) | | 0.046 (0.043 0.048) | | |
| Prob. RMSEA < = .05 | 0.000 | | 0.999 | | |
| CFI | 0.671 | | 0.929 | | |
| SRMR | 0.082 | | 0.033 | | |

Note: All coefficients are significant, $n$ = 7,241.

[a]factor loading.

[b]social distance.

All the β-coefficients or factor loadings are 0.4 or higher and statistically significant, which is required for good construct validity. Finally, given that the correlation between the generalized trust and prejudice factors is relatively low, generalized trust constitutes a separate latent factor. This is important because it implies that the friendship-prejudice results from the social-psychological literature [26–28] need not spill over into the friendship-trust relation. As such, we cannot simply *assume* that there is a substantively meaningful friendship-trust relation, and a more detailed analysis of this relation is in order. We turn to this analysis in the next section.

## Structural equation modelling results

Our analysis of the friendship-trust relation relies on two structural models. Both were identified and could be tested. While the generalized trust and prejudice latent factors retrieved above will be included in the first model, the second model also includes threat and negative affect latent scales as mediators. The first model tests whether there is a total effect of friendship on generalized trust and prejudice, controlling for socio-demographics, socio-economic factors, and country effects. The result of this structural model is depicted in Fig 2 below. As we can see from the fit statistics reported at the bottom of Fig 2, the fit is moderate. The model also indicates a statistically significant *total* effect of cross-group friendship on generalized trust. Yet, this is substantially weaker than the negative *total* effect of cross-group friendship on prejudice. The 99% BC CI of its standardized value lies between 0.012 and 0.106. For the total effect of cross-group friendship on prejudice the 99% BC CI lies between -0.362 and -0.279. Since the highest absolute value of the path between cross-group friendship to trust and the lowest absolute value of the path from cross-group friendship to prejudice do not overlap, we can be confident that these estimated values are substantially and statistically different.

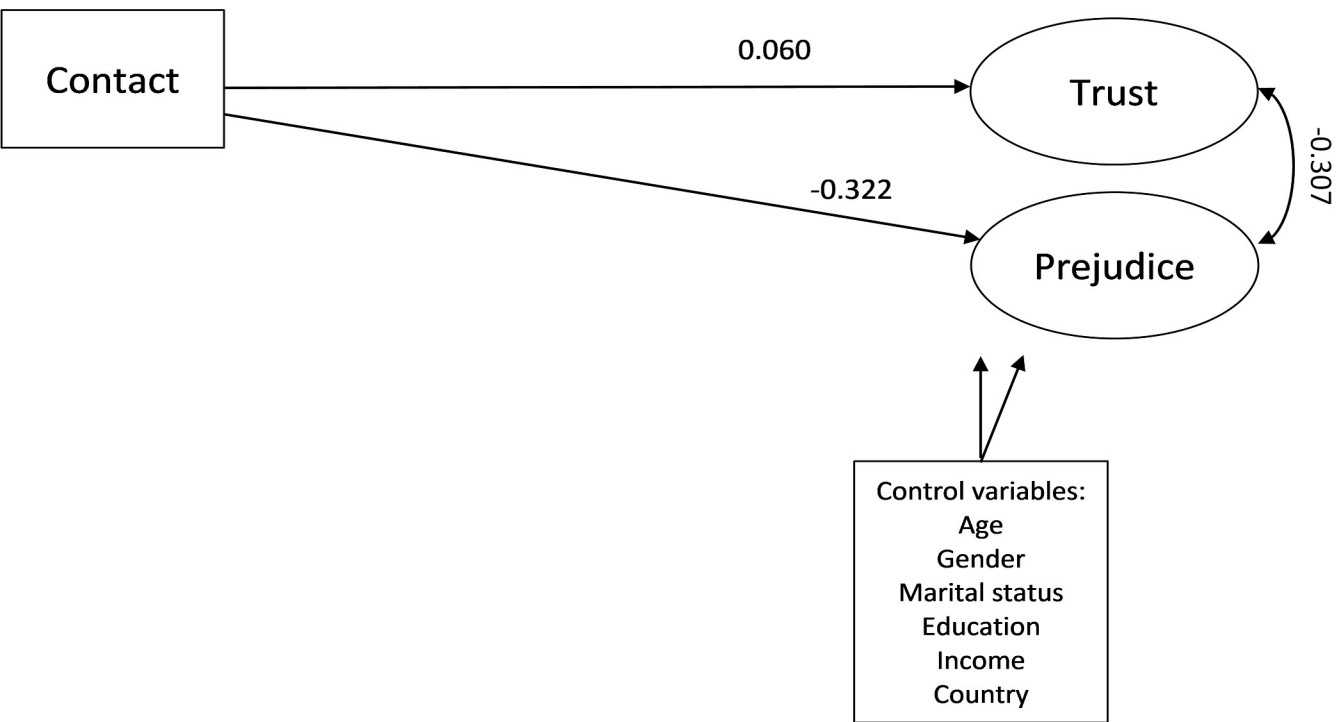

**Fig 2. Structural model of generalized trust and prejudice.** Model fit indices: *RMSEA* = 0.060; *Prob. RMSEA* < = .05: 0.00 (*CI*: 0.057–0.062); *CFI* = 0.773; *SRMR* = 0.057; *n* = 5,333.—: Significant paths.

Therefore, we refute $H_{1A}$ (an equally strong total effect), in favour of $H_{1B}$ (comparatively weaker total effect of cross-group friendship on generalized trust).

Fig 3 includes the two potential mediators (i.e. negative affect and threat perceptions), which allows assessing both their direct effects on the dependent variables, as well as how they mediate any indirect effects. The model illustrates that lower levels of negative affect and threat are predicted by cross-group friendship. Negative affect only statistically significantly relates to cross-group friendship, but not to generalized trust. However, threat is significantly negatively related to generalized trust, and it is worth pointing out here that the effect size of this path (β-coefficient) is substantive (-0.290). We discuss this result and the estimated size of the indirect effects later. For now it is important to note that the direct effect of cross-group friendship on generalized trust becomes rather weaker (-0.013) and statistically insignificant once mediators are included. Unsurprisingly, the direct effect of cross-group friendship on prejudice decreases, but remains statistically significant (compare -0.322 in Fig 2 to -0.160 in Fig 3). Finally, we should note that the model fit is very well as indicated by a RMSEA value of below 0.05, while a CFI value that approximates 0.90 demonstrates a reasonable fit.

To cross-validate our results with the standard finding in social-psychological research on contact and prejudice, the model above also includes the mediated effect of threat and negative affect on prejudice (see Fig 3 again). The indirect effect of cross-group friendship on prejudice runs through both threat and negative affect. Lower levels of negative affect and threat statistically significantly impact prejudiced feelings. Our results thus relate well to experimental findings on the contact-prejudice relation in social psychology [26–28]. More importantly, however, both mediators also show significant direct effects on prejudice. Especially the direct

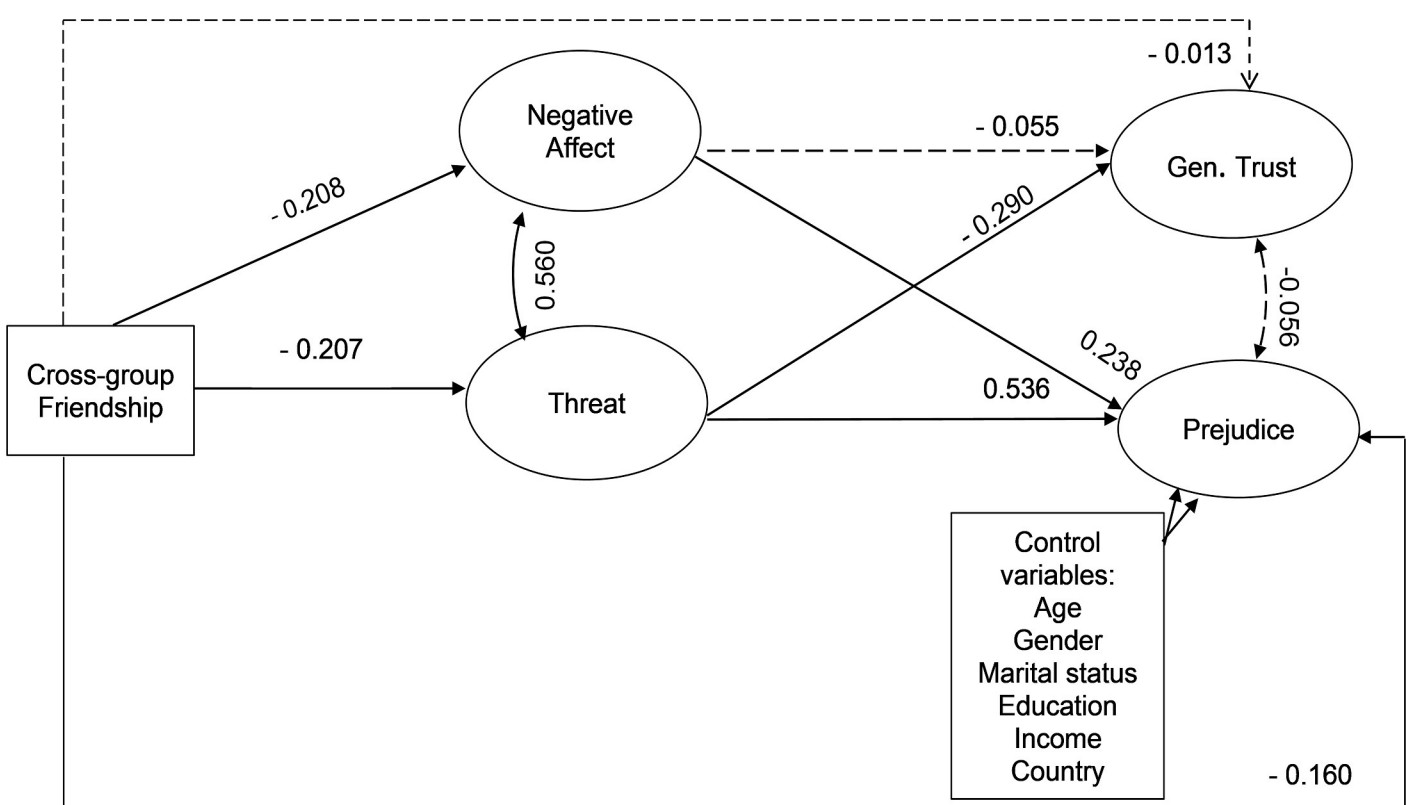

**Fig 3. Structural model of generalized trust and prejudice with mediators.** Model fit indices: *RMSEA* = 0.043; *Prob. RMSEA< = .05*: 1.00 (*CI*: 0.041–0.044); *CFI* = 0.857; *SRMR* = 0.091; *n* = 5,333. —: Non-significant paths.—: Significant paths.

**Table 3. Total, direct and indirect effects on generalized trust and prejudice, standardized.**

| Predictor | Mediator | Dependent variable | Total | Total indirect | Direct | Indirect |
|---|---|---|---|---|---|---|
| Cross-group Friendship | | Generalized Trust | 0.059 (0.012, 0.106) | 0.072 (0.052, 0.092) | -0.013 (n.s.) | — |
| | Negative Affect | | | | -0.055 (n.s.) | 0.012 (n.s.) (-0.011, 0.034) |
| | Threat | | | | -0.290 | 0.060 (0.040, 0.080) |
| Cross-group Friendship | | Prejudice | -0.321 (-0.362, -0.279) | -0.161 (-0.192, -0.129) | -0.160 | — |
| | Negative Affect | | | | 0.238 | -0.050 (-0.074, -0.025) |
| | Threat | | | | 0.536 | -0.111 (-0.137, -0.085) |

Note: n.s.: Not significant; the total effects reported here are marginally different than the effects in Fig 2 due to rounding. In brackets the 99% BC CI are reported for effects that contain products of two terms.

effect of threat on prejudice is substantial (0.536). The direct effect of negative affect is lower (0.238) and remains statistically significant. This suggests that cross-group friendship not only has significant direct effects on prejudice, but that there are also substantial indirect effects running through both mediators.

Table 3 summarises the relative size of total, total indirect as well as direct and indirect effects of cross-group friendship on generalized trust and prejudice (some of which were previously presented in Figs 2 and 3). When we assess the total indirect effects of friendship through the mediators on generalized trust and prejudice we see that the latter is comparatively much stronger (-0.161 versus 0.072). The indirect path between friendship and generalized trust mediated by negative affect is substantially weaker than that mediated by threat (respectively 0.012 and 0.060) and is not statistically significant. In addition, the indirect effect of friendship on trust through threat is much weaker (0.060) than that on prejudice (-0.111). If we evaluate the absolute values at the lower and upper bounds of the 99% BC CIs of these effect sizes, we see that these do not overlap (0.040–0.080 versus -0.137 –-0.085). We can thus be confident that these effects statistically and significantly differ in size. Overall, the evidence in favour of $H_{2A}$ is limited, and we refute $H_{2A}$ (equally strong total indirect effects), in favour of $H_{2B}$ (a weaker total indirect effect of cross-group friendship on generalized trust than that of cross-group friendship on prejudice). These results indicate that even by including a necessary condition of intergroup relations theory (cross-group friendship) as well as the presumed mediators in a large multi-country analysis controlling for many factors, much of the variance in generalized trust remains unexplained. Still, the direct path between threat and generalized trust (-0.290), which is one of our highest effect sizes, is in line with prior research [5, 63, 64]. We will now turn to discussing the implications of these results.

## Discussion and conclusion

Despite the crucial role often attributed to intergroup contact when discussing the relationship between ethnic diversity and generalized trust, direct tests of a friendship-trust relation have remained rare at the individual level [2, 3]. In this article, we have built on the friendship-prejudice connection and its mediators studied in social-psychological research [15, 26–28] to study the relation between cross-group friendship and generalized trust in more detail. Our effort could thereby be seen as an answer to Pettigrew's [65] call for multi-country, interdisciplinary and integrative analyses of intergroup relations.

Using a number of structural equation models, we observe that the *total* effect of cross-group friendship on generalized trust is statistically significant, but rather weak. There are also statistically significant *indirect* effects of cross-group friendship on generalized trust, which run primarily via perceptions of threat. The mediating role of negative affect could not be

supported in this context. Overall, cross-group friendship thus appears to relate to lower levels of perceptions of threat, and such perceptions of threat are associated with lower levels of generalized trust. Yet, these indirect effects are weaker than those observed in paths with prejudice as an outcome variable. Therefore, the message appears to be that threat perceptions and negative affect are a fundamental part of the friendship-prejudice relation, but much less so for the friendship-trust relation. As such, our results provide important new insights about the mechanisms determining generalized trust, and may help clarify why earlier aggregate-level research has mostly found weak or inconsistent findings on the friendship-trust relation. In addition, our approach highlights the need to pay more attention to substantive relationships between indicators of social cohesion rather a sole emphasis on statistical association [4, 66].

Despite these contributions, our study has also some limitations. One persistent and unaddressed concern relates to a potential endogeneity problem. Our dataset is indeed unable to establish (the direction of) causal chains with certainty. While such endogeneity and causality concerns can be overcome via panel data, the small effect sizes observed in our analysis suggest that we may need frequent measurements over a lengthy period to decompose time varying (macro and micro), and time invariant (i.e. individual-level fixed) effects. For trust, such a panel will be a costly endeavour with potentially unsubstantial effect sizes suggested by our results. As an alternative approach, one could consider field experiments, which allow for much stronger causal inferences. A few recent studies have taken this route to study the relation between personal contact and anti-immigrant sentiments as well as trust [67, 68].

Another limitation is that we cannot directly model contextual diversity in residential settings, which may shape opportunities for intergroup contact [69–73]. While future work could consider extending the current structural model with measures of contextual diversity, we believe that this does not affect the conclusions of this study [39]. The reason is that our study evaluates the (in)direct effect(s) of cross-group friendship, *whatever has led to its establishment*. Once cross-group friendship is in place, our results suggest that it relates negatively to threat perceptions whereas it only shows a minor association with generalized trust. One reason why we have found such a weak *total effect* of cross-group friendship on generalized trust and a weak *total indirect effect*, is that their influence may be confounded by other, unmeasured variables such as community disadvantage [4, 74] or negative contact [75]. This is an important avenue for further research.

Finally, it is possible that our (weak) findings are affected by the operationalization of our key concepts. For example, our prejudice scale includes an item on (un)willingness to send one's children to a majority immigrant school, which could be based on reasonable concerns about the lack of resources rather than inflexible generalizations about out-groups. While a common concern for all empirical research into social and psychological concepts, we should note that we follow standard operationalizations for both prejudice and trust. Even though these operationalizations may be imperfect, this implies that we are capturing the same 'prejudice' and 'trust' (whether or not combined with other things) as in previous scholarship. Still, robustness checks using alternative operationalizations would be important to substantiate our findings.

In sum, the relationship between cross-group friendship and generalized trust is in our analysis found to be less strong than that between cross-group friendship and prejudice. When studying generalized trust at the individual level, we therefore suggest analysts should consider alternative mechanisms. Social psychology has in intergroup relations theory one of its most valuable explanatory tools. Despite its intuitive appeal, scholars and policymakers alike should not lose sight of areas in which it may have little predictive power. The present study articulates this problem for individual-level studies of generalized trust that invoke intergroup relations theory.

## Supporting information

**S1 File.**
(ZIP)

## Acknowledgments

The authors gratefully acknowledge valuable comments of panel members at the American Political Science Association 2013 conference (organizer: Yoshika M. Herrera, chair: Susan J. Pharr, discussant: Donna Bahry) and the two anonymous reviewers.

## Author Contributions

**Conceptualization:** Wahideh Achbari.

**Formal analysis:** Wahideh Achbari.

**Funding acquisition:** Wahideh Achbari, Benny Geys.

**Methodology:** Wahideh Achbari.

**Resources:** Bertjan Doosje.

**Writing – original draft:** Wahideh Achbari, Benny Geys.

**Writing – review & editing:** Wahideh Achbari, Benny Geys, Bertjan Doosje.

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
