## [Decision Letter · Decision Letter 0]

1 Jun 2020

PONE-D-20-11120

Comparing the effect of cross-group friendship on generalized trust to its effect on prejudice: The mediating role of threat perceptions and negative affect

PLOS ONE

Dear Dr. Achbari,

Thank you for submitting your manuscript to PLOS ONE. After careful consideration, we feel that it has merit but does not fully meet PLOS ONE’s publication criteria as it currently stands. Therefore, we invite you to submit a revised version of the manuscript that addresses the points raised during the review process.

I have solicited advice from two expert reviewers, who have returned the reports shown below. Both reviewers are generally positive about the paper and recommend revisions. However, they also raise a number of important issues that need to be resolved before the paper can be publishable. Reviewer #1 is mostly concerned about the interpretation of the statistical analysis, most specifically about the discussion on effect sizes. You should take this point very seriously, in addition to addressing all the small points raised by the reviewer. Reviewer #2 is somewhat more positive about the paper; however, there are some suggestions about the conceptual part of the paper that can sharpen the argument and improve the legibility of the paper. They should be incorporated in the revised version.

We look forward to receiving your revised manuscript.

Kind regards,

Luis M. Miller, Ph.D.

Academic Editor

PLOS ONE

Reviewers' comments:

Reviewer's Responses to Questions

**Comments to the Author**

1. Is the manuscript technically sound, and do the data support the conclusions?

Reviewer #1: Partly

Reviewer #2: Partly

2. Has the statistical analysis been performed appropriately and rigorously? 

Reviewer #1: I Don't Know

Reviewer #2: Yes

3. Have the authors made all data underlying the findings in their manuscript fully available?

Reviewer #1: Yes

Reviewer #2: Yes

4. Is the manuscript presented in an intelligible fashion and written in standard English?

Reviewer #1: Yes

Reviewer #2: Yes

5. Review Comments to the Author

Reviewer #1: Review of “Comparing the effect of cross-group friendship on generalized trust to its effect on prejudice: The mediating role of threat perceptions and negative affect”

This study investigates the role of cross-group friendship in reducing the negative effects of

ethnic diversity on generalized trust. Particularly, they aim to fill a gap in previous empirical research that has overlooked the hypothesized (and often assumed) direct effect of cross-friendships in generalized trust. The authors compare the effect sizes of cross-friendships in generalized trust to the effect on prejudice, which has been empirically tested and corroborated before. They also empirically test whether generalized trust is affected by cross-group friendship via lower levels of threat and negative affect (perceived conflict) at the individual level, which are central assumptions of research in social psychology.

Comments

This study investigates a topic of great importance for both the literature in social psychology and political and social interventions to increase ethnic integration. A topic that is especially relevant today as societies are growing in diversity. The topic, however, has been extensively researched before, which makes contributing to the field a bit hard. However, the authors were able to see a gap in the literature and tried to address it. That said, I think the paper has a few major shortfalls that should be addressed before publication. These shortfalls contradict PlosOne publication standards. My main 2 concerns are the following:

1. Conclusions are not supported by the data: In the hypotheses you want to test different “effect sizes”, however, the statistical analyses do not focus on this issue, but rather on coefficient values (betas) and their p-value. P-values are not a measurement of the effect size and the beta coefficients can only be a measure when certain conditions are met. Furthermore, the authors claim to demonstrate that the effect can be negligible. I would suggest the authors to either delete the references to effect size in the hypotheses or present analyses on the effect sized and the nature of the null effect. The authors could perhaps run a equivalence test for the null effect. If I am getting this wrong, I am, of course open to read the author’s counter-argument. As you will see in my specific comments, I think the authors should tone down the implications of the findings, and discuss more in-depth the limitations of the analyses they perform. Are there any limitations in their data analysis that may explain the lack of link between cross-friendship and generalized trust?

2. The paper is written in a way that it is somewhat difficult to understand in some parts. The authors use complicated rhetoric and repeat information, often superfluous, a lot. I am not a native speaker myself so I am not talking about the quality of the language per se (which is fine in my opinion), but the clarity of the exposition. I would suggest the authors work a bit on the literature review section and focus on the research gap they intent to address because the goal of the paper or how the authors arrive to conclusions is not always clear. The results and discussion section are ok, although the discussion is a bit too long.

I made specific some comments to the authors that they may find interesting to improve the paper.

Introduction and literature review

• The authors make a very good job at explaining and describing the research process: how and what their study will test. However, it comes across as somewhat disorganized and too technical for an introduction. I was expecting to be told why I should be interested in the topic, how the findings will fit in the literature. For example, in the last paragraph in page 3, the authors first explain that they intend to conduct a confirmatory factor analysis and then they finish the paragraph explaining what we can learn from such analysis. I would suggest the authors that they reverse this order and first state what they want to learn and why, and only then explain how they plan to do it.

• Page 4: “Sufficient studies confirm…” Sufficient is something to be judged by the audience so I would recommend the authors to write something that refers just to the amount such as a lot, few, many studies…

• Page 5, line 98: “stimulating positive subjective orientations towards”. This is maybe a very well formulated description for a psychology audience but PlosOne is a journal targeting a general audience so I would recommend the authors to reduce the use of jargon. It makes the paper quite difficult to follow.

• Page 5: the authors discuss that cross-group friendships meet all Allport’s conditions to reduce prejudice. I think this fact should be introduced earlier.

Hypotheses

• Can the authors elaborate on why they compare the 2 causal effects sizes? Why is the comparison of the size effect interesting? What would imply a disparity of the effect sizes? If the direct effect of cross-friendship in trust has not been yet tested as the authors claim? Shouldn’t they test this first?

Results

• In page 17, the authors conclude that “The model also indicates a weak and statistically significant total effect of cross-group friendship on generalized trust and a relatively substantial negative total effect of cross-group friendship on prejudice. Therefore, we can refute H1A (an equally strong total effect), but not H1B (comparatively weaker total effect of cross-group friendship on generalized trust). Both these paths 334 will next be examined including the mediators in the models.” How do you measure effect sized? Does you analytical strategy actually test these hypotheses?

• I do not understand the distinctions made by the authors between direct and indirect effects. Wouldn’t all effects bi direct until the mediators are introduced in the model? My knowledge of mediation analysis in very limited so I would appreciate if the authors could elaborate in this topic. The explanation doesn’t need to go in the paper.

Conclusion

• Page 21 lines 400-401, the authors state “our findings demonstrate that the total effect of cross-group friendship on generalized trust is statistically significant, but rather weak.” I believe the authors overstate the meaning of their findings. First, I would like the authors to tone down a bit the statements, 2) the authors should distinguish between “we didn’t find an effect” and “there is no effect”, 3) explain what could have failed (is the operationalization of the concept maybe affecting the results in any way?), and 4) perhaps conducting equivalence testing on the effect would be a nice idea? That way, the authors could argue that the effect is not larger than the minimum effect of interest. This is just a suggestion. The authors can tell me whether they think this is a good idea.

• In the conclusion, the authors devote an extensive part to the limitations. While I do think that the limitations should be highlighted, it’s be better if the authors could say how the limitations affected the findings and how or whether these limitations may be overcome. Particularly, I have trouble with the paragraph on the direction of causality as this is contradicting the section of the direction of causality. Should I then trust previous literature in this or not? What efforts have you made, if any, to overcome this general limitation?

• Line 456: The authors again use the work “demonstrate” and I would suggest hem to tone down this a little bit or perform analyses that can support this asseveration.

• Last paragraph: “As we demonstrate here, the relationship between cross-group friendship and generalized trust is less strong than that between cross-group friendship and prejudice. When studying generalized trust at the individual level, we therefore suggest analysts should consider alternative mechanisms…” Again, demonstrate is too strong. However, this implication is in fact the most interesting.

• Are the scripts for data analyses available?

Reviewer #2: This is an interesting paper that studies an important and under-analyzed topic, the role of cross-group friendship on trust and prejudice. In general terms, I like it, although I think that the theory is under-developed and the results are probably driven to a certain extent by the indicators of prejudice and trust used in the paper. These are my comments:

- The idea is that cross-group friendship may enhance a broader willingness to trust in others. But why trust in general others? Cross-group friendship could have an effect on trust in the ethnic group of your friend, so, perhaps, we should see an effect of cross-group friendship on out-group trust, but not so much on social trust. Friendship clearly positively affects your beliefs about your friend’s trustworthiness (actually, friendship and trust are closely related, as Cicero would say). It is not so clear to what extent it affects your beliefs about general others. Clearly, your friends are not a representative sample of society, so in principle, friendship would affect just particularized forms of trust. Moreover, if the cross-group friendship is towards a member of a minority ethnic group, even if the truster considers that her friend is somewhat representative of her ethnic group, this may not have much force in his general assessment of the distribution of types in the wider society. What I mean is that his assessment of the distribution of types in society could be to a big extent disconnected of his beliefs about the trustworthiness of a specific ethnic group.

- I have some problems disentangling trust from prejudice. In what sense these beliefs can be differentiated from the belief about someone’s trustworthiness? Some of the reasons for not being willing to send children to a school with a majority of immigrants, or to move to a district with many immigrants may be based on beliefs neither related to prejudice or to trust. For example, you may be reluctant to send your children to schools with a majority of immigrants because you have seen (in official reports, perhaps) that the quality of the education in these schools is relatively low, or you may not want to move to a district with many immigrants because you fear that this would affect to the future price of your home (perhaps not because of your prejudice but because of other people's prejudice against immigrants). Likewise, you can be in favour of giving the employer the right to only employ nationals because, as a national, it benefits you. But to the extent that these indicators do capture prejudices, in what sense do they differ from trust? It is remarkable that there is no definition of what prejudice is in the paper (neither of what trust is, for that matter). But if prejudice is the notion that someone is dishonest, or do not play by the rules of the host country, how is this different from the belief on someone’s trustworthiness? I wonder whether the results can be partly explained because what the authors are measuring is not actually prejudice, but something that combines prejudice (which, as I said, is a belief that should be very close to trusting) and other things.

- The authors dismiss the problem of endogeneity in a rather cavalier manner; it may be true that if the results refute the hypothesis, there is no point in conducting research with a stronger analysis; but they have to be conscious that with cross-sectional data and not causality test (such as an instrumental variable approach, for example), the results just say that it could exist a causal relationship going from cross-group friendship to prejudice, but it could also be the other way round: less prejudiced people are more likely to have more friends from other ethnic groups.

- A minor point, the authors claim that cross-group friendship meets the condition of equal social status: Is this the case when people have a different ethnic background? Has a member of the minority group the same social status than the member of the majority?

6. PLOS authors have the option to publish the peer review history of their article (what does this mean?). If published, this will include your full peer review and any attached files.

Reviewer #1: No

Reviewer #2: No

---

## [Author Response · Author response to Decision Letter 0]

26 Oct 2020

Reviewer #1: 

This study investigates a topic of great importance for both the literature in social psychology and political and social interventions to increase ethnic integration. A topic that is especially relevant today as societies are growing in diversity. The topic, however, has been extensively researched before, which makes contributing to the field a bit hard. However, the authors were able to see a gap in the literature and tried to address it. That said, I think the paper has a few major shortfalls that should be addressed before publication. 

Thank you for your positive evaluation of the importance of our topic and the recognition of the contribution of our manuscript. We have taken your constructive suggestions for further improvements to heart, and set out below how we have addressed your comments.

1. Conclusions are not supported by the data: In the hypotheses you want to test different “effect sizes”, however, the statistical analyses do not focus on this issue, but rather on coefficient values (betas) and their p-value. P-values are not a measurement of the effect size and the beta coefficients can only be a measure when certain conditions are met. Furthermore, the authors claim to demonstrate that the effect can be negligible. I would suggest the authors to either delete the references to effect size in the hypotheses or present analyses on the effect sized and the nature of the null effect. The authors could perhaps run a equivalence test for the null effect. If I am getting this wrong, I am, of course open to read the author’s counter-argument. As you will see in my specific comments, I think the authors should tone down the implications of the findings, and discuss more in-depth the limitations of the analyses they perform. Are there any limitations in their data analysis that may explain the lack of link between cross-friendship and generalized trust?

We thank the reviewer for pushing us to clarify our approach and agree that coefficients and p-values are not sufficient to claim that our effects are large enough or practically important/significant. Indeed, trivially small effects can be statistically significant with large sample sizes. Although significance does carry information, evaluating mediation should not solely be defined in terms of statistical significance. We also agree that equivalence tests can be employed to assess whether an effect is small enough to be negligible. However, the information on which such confidence intervals are based has to be derived from external sources (such as arbitrary rules of thumb, census information, prior research, or when in conjunction with power analyses, derived from a feasible sample size; (Lakens, Scheel, and Isager 2018)). In our case, there is unfortunately very little prior research on the total effect of cross-group friendship on trust, and no prior study engaged in a mediation analysis. Therefore, we decided to assess our findings against the benchmark of an established path in social psychological literature, namely that between cross-group friendship (x), negative affect and threat (z), and prejudice (y). In practice, this means that we evaluate our effect size(s) in terms of a specified value, which is i) theoretically based on prior research in intergroup relations, and ii) simultaneously estimated for the same respondent using the same data source. Note that this approach follows the definition of effect size by Preacher and Kelly (2011: 95, our italics) as: “any measure that reflects a quantity of interest, either in an absolute sense or as compared with some specified value”. “Standardized effect sizes are on a meaningful scale in units of standard deviations”. McKinnon (2008: 81-85) similarly recommends evaluating standardized effect sizes of individual and mediated paths. Moreover, it is important to note that Betas derived from Structural Equation Modeling are often deemed superior since these are simultaneously estimated with multiple variables and assessed in light of measurement error (a latent variable approach) (Hancock 2001). We do discuss magnitudes in relation to each other since this is imperative to the goal of the paper, but have followed your recommendations and in the literature (Preacher and Kelley 2011), which suggest avoiding providing excessive commentary about the sizes and removed the word “negligible” in this regard throughout the manuscript.

Nevertheless, we agree that we should formally test whether our Betas statistically differ rather than eyeballing their magnitude. There are several tests available for the equality of effect sizes, such as the Wald confidence interval (CI) through Sobel’s method, likelihood-based CI, and variations of Bootstrap CI (Cheung 2007). The Bootstrap CI outperforms all since it does not assume that the data are multivariate normal, is not sensitive to sample size, and it is not symmetric around the estimated mediating effect. The latter is especially important since “the sampling distribution of a mediation effect is complicated because the mediation effect is quantified by a product of at least two parameters” (Ryu and Cheong 2017: 1). More specifically, “the sampling distribution of the product term ab is not normal … even though a and b are normally distributed” (Cheung 2007: 231; MacKinnon 2008: 95). That is to say, even if two effects are statistically significant their joint test is not necessarily so. For these reasons comparing Bootstrap CI and whether these would not overlap is superior to other tests. We have conducted such tests and reported these CIs, and discuss whether these do not overlap for our effect sizes of interest in the text. The results are in line with what we had reported before, which is due to the observed large differences.

“In addition, we report Bias corrected Bootstrapped Confidence Intervals (2000 iterations) for our total, total indirect, and indirect effects in order to compare the effect sizes. The Bootstrapped Confidence Intervals (BC CI) outperforms other tests since it does not assume that the data are multivariate normal, is not sensitive to sample size, and it is not symmetric around the estimated mediating effect. The latter is important because even if two effects are statistically significant their joint test is not necessarily so [57 p. 231,58 p. 95,59 p. 1].” (p. 14)

“The model also indicates a statistically significant total effect of cross-group friendship on generalized trust. Yet, this is substantially weaker than the negative total effect of cross-group friendship on prejudice. The 99% BC CI of its standardized value lies between 0.013 and 0.106. For the total effect of cross-group friendship on prejudice the 99% BC CI lies between -0.363 and -0.280. Since the highest absolute value of the path between cross-group friendship to trust and the lowest absolute value of the path from cross-group friendship to prejudice do not overlap, we can be confident that these estimated values are substantially and statistically different.” (p. 17)

“the indirect effect of friendship on trust through threat is much weaker (0.060) than that on prejudice (-0.111). If we evaluate the absolute values at the lower and upper bounds of the 99% BC CIs of these effect sizes, we see that these do not overlap (0.040 – 0.080 versus -0.137 – -0.085). We can thus be confident that these effects statistically and significantly differ in size.” (pp. 19-20)

In the previous version two of the control variables had been mistakenly omitted in parts of the model. The inclusion of these variables introduced very minor changes in the model fits and coefficients. In addition, in the most recent version of Mplus 8, instead of WRMR, SRMR is reported as one of the Goodness of Fit indicators, which we now have followed. 

2. The paper is written in a way that it is somewhat difficult to understand in some parts. The authors use complicated rhetoric and repeat information, often superfluous, a lot. I am not a native speaker myself so I am not talking about the quality of the language per se (which is fine in my opinion), but the clarity of the exposition. I would suggest the authors work a bit on the literature review section and focus on the research gap they intent to address because the goal of the paper or how the authors arrive to conclusions is not always clear. The results and discussion section are ok, although the discussion is a bit too long.

Thank you for pointing this out. During the revision process, we have carefully edited the manuscript to clarify and simplify our exposition, remove superfluous repetitions and streamline as well as shorten the discussion section. We have thereby also followed the more detailed suggestions you made regarding the separate sections of our manuscript, and are grateful to you for pushing us further in this direction.

Introduction and literature review

• The authors make a very good job at explaining and describing the research process: how and what their study will test. However, it comes across as somewhat disorganized and too technical for an introduction. I was expecting to be told why I should be interested in the topic, how the findings will fit in the literature. For example, in the last paragraph in page 3, the authors first explain that they intend to conduct a confirmatory factor analysis and then they finish the paragraph explaining what we can learn from such analysis. I would suggest the authors that they reverse this order and first state what they want to learn and why, and only then explain how they plan to do it.

• Page 4: “Sufficient studies confirm…” Sufficient is something to be judged by the audience so I would recommend the authors to write something that refers just to the amount such as a lot, few, many studies…

• Page 5, line 98: “stimulating positive subjective orientations towards”. This is maybe a very well formulated description for a psychology audience but PlosOne is a journal targeting a general audience so I would recommend the authors to reduce the use of jargon. It makes the paper quite difficult to follow.

• Page 5: the authors discuss that cross-group friendships meet all Allport’s conditions to reduce prejudice. I think this fact should be introduced earlier.

We have followed each of these suggestions to simplify and clarify our introduction.

Hypotheses

• Can the authors elaborate on why they compare the 2 causal effects sizes? Why is the comparison of the size effect interesting? What would imply a disparity of the effect sizes? If the direct effect of cross-friendship in trust has not been yet tested as the authors claim? Shouldn’t they test this first?

Our primary interest lies in the total, direct, and indirect effects of cross-group friendship on trust. Yet, as mentioned in relation to your first comment, we use the results on prejudice as a benchmark to assess our estimated effect sizes. The comparison between the observed effects on trust and prejudice thus is important to present and discuss since it allows us to embed our findings more closely in the existing literature (which studied prejudice, but not yet trust). 

Results

• In page 17, the authors conclude that “The model also indicates a weak and statistically significant total effect of cross-group friendship on generalized trust and a relatively substantial negative total effect of cross-group friendship on prejudice. Therefore, we can refute H1A (an equally strong total effect), but not H1B (comparatively weaker total effect of cross-group friendship on generalized trust). Both these paths will next be examined including the mediators in the models.” How do you measure effect size? Does you analytical strategy actually test these hypotheses?

Mediation can be defined in terms of the following equation c = c’ + a*b (in this example there is only one y-variable), whereby c stands for the total effect, c’ for the direct effect, and a*b for the indirect effect. Our hypotheses H1A and H1B test for the equality of the magnitude of c (the total effects). Our hypotheses H2A and H2B test for the equality of the magnitude of the indirect effects. Note that in our case there are multiple y-variables and mediators present, the effects of which are added.

Effect size was interpreted in terms of the estimated standardized coefficients, although we agree with your earlier comment that there are important assumptions underlying the validity of this approach. That being said, an important insight can be obtained from observing in our analysis that cross-group friendship is associated with lower levels of prejudice in a substantial and statistically significant manner (as also shown in previous work), while it only has a small effect on generalized trust (a new finding from our analysis).

• I do not understand the distinctions made by the authors between direct and indirect effects. Wouldn’t all effects be direct until the mediators are introduced in the model? My knowledge of mediation analysis in very limited so I would appreciate if the authors could elaborate in this topic. The explanation doesn’t need to go in the paper.

It is true that in a model without mediators the total effect of variable X on variable Y and the direct effect of variable X on variable Y will be the same. In such a setting, there thus is perfect equivalence between the ‘total’ and ‘direct’ effects, and we then would abstain from adding the word ‘total’ or ‘direct’. Yet, once mediators are introduced, this is no longer true. Then it becomes possible to separate the total effect of variable X on variable Y into a direct effect of variable X on variable Y and an indirect effect of variable X on variable Y via variable Z. The total effect then is defined as the sum of all direct and indirect effects linking the variables X and Y (Sobel 1987).

Conclusion

• Page 21 lines 400-401, the authors state “our findings demonstrate that the total effect of cross-group friendship on generalized trust is statistically significant, but rather weak.” I believe the authors overstate the meaning of their findings. First, I would like the authors to tone down a bit the statements, 2) the authors should distinguish between “we didn’t find an effect” and “there is no effect”, 3) explain what could have failed (is the operationalization of the concept maybe affecting the results in any way?), and 4) perhaps conducting equivalence testing on the effect would be a nice idea? That way, the authors could argue that the effect is not larger than the minimum effect of interest. This is just a suggestion. The authors can tell me whether they think this is a good idea.

We agree that there is an important distinction between “we didn’t find an effect” and “there is no effect”, and we have been more precise in our wording in the revised manuscript. We thereby also toned down the language. It is indeed perfectly possible that we failed to pick up stronger effects due to the operationalization of our concepts. Still, we should note in this respect that we have followed standard operationalizations for both prejudice and trust, such that our findings are based on the state-of-the-art in previous work. We have been more explicit about this in the revised version of the manuscript.

“Using a number of structural equation models, we observe that the total effect of cross-group friendship on generalized trust is statistically significant, but rather weak. There are also statistically significant indirect effects of cross-group friendship on generalized trust, which run primarily via perceptions of threat. The mediating role of negative affect could not be supported in this context. Overall, cross-group friendship thus appears to relate to lower levels of perceptions of threat, and such perceptions of threat are associated with lower levels of generalized trust.” (p. 20)

“Finally, it is possible that our (weak) findings are affected by the operationalization of our key concepts. While a common concerns for all empirical research into social and psychological concepts, we should note that we follow standard operationalizations for both prejudice and trust. Even though these operationalizations may be imperfect, this implies that we are capturing the same ‘prejudice’ and ‘trust’ (whether or not combined with other things) as in previous scholarship. Still, robustness check using alternative operationalizations would be important to substantiate our findings.” (p. 22)

• In the conclusion, the authors devote an extensive part to the limitations. While I do think that the limitations should be highlighted, it’s be better if the authors could say how the limitations affected the findings and how or whether these limitations may be overcome. Particularly, I have trouble with the paragraph on the direction of causality as this is contradicting the section of the direction of causality. Should I then trust previous literature in this or not? What efforts have you made, if any, to overcome this general limitation?

Thank you for pointing this out. We have rephrased our limitations section to indicate how these limitations might affect the results presented in the manuscript. We have also revised our paragraph on the direction of causality in the conclusion to make it consistent with the discussion of this same issue earlier in the manuscript. In our view, previous work on the implied direction of causality can be trusted, but overcoming this limitation would require longitudinal data with exogenous variation in cross-groups friendships (i.e. unrelated to self-selection). A few recent studies based on experimental research designs have taken steps in this direction (Finseraas and Kotsadam, 2017; Finseraas et al., 2019), but such data was unfortunately unavailable to us. We have discussed this in the revised version of our concluding discussion.

• Line 456: The authors again use the work “demonstrate” and I would suggest hem to tone down this a little bit or perform analyses that can support this asseveration.

• Last paragraph: “As we demonstrate here, the relationship between cross-group friendship and generalized trust is less strong than that between cross-group friendship and prejudice. When studying generalized trust at the individual level, we therefore suggest analysts should consider alternative mechanisms…” Again, demonstrate is too strong. However, this implication is in fact the most interesting.

We have toned down the language and avoid the use of the word ‘demonstrate’ (or equivalents).

• Are the scripts for data analyses available?

Yes. All scripts will be made available. 

Reviewer #2: 

This is an interesting paper that studies an important and under-analyzed topic, the role of cross-group friendship on trust and prejudice. In general terms, I like it, although I think that the theory is under-developed and the results are probably driven to a certain extent by the indicators of prejudice and trust used in the paper.

Thank you for your positive evaluation of our manuscript, and your constructive suggestions for further improvements. We set out below how we have addressed your comments.

- The idea is that cross-group friendship may enhance a broader willingness to trust in others. But why trust in general others? Cross-group friendship could have an effect on trust in the ethnic group of your friend, so, perhaps, we should see an effect of cross-group friendship on out-group trust, but not so much on social trust. Friendship clearly positively affects your beliefs about your friend’s trustworthiness (actually, friendship and trust are closely related, as Cicero would say). It is not so clear to what extent it affects your beliefs about general others. Clearly, your friends are not a representative sample of society, so in principle, friendship would affect just particularized forms of trust. Moreover, if the cross-group friendship is towards a member of a minority ethnic group, even if the truster considers that her friend is somewhat representative of her ethnic group, this may not have much force in his general assessment of the distribution of types in the wider society. What I mean is that his assessment of the distribution of types in society could be to a big extent disconnected of his beliefs about the trustworthiness of a specific ethnic group.

We agree that one would first of all expect to see an effect of cross-group friendship on out-group trust. However, our argument is that such effects – once established – might then spill over to broader-based social trust via so-called secondary transfers. We have been more detailed about the underlying theoretical argument in the revised version of the manuscript. Clearly, however, whether or not such hypothesized effect on social trust materializes remains an empirical question, which we address in our analysis.

“Generalized trust not only requires reduced prejudice or lower levels of negative emotions towards a specific out-group. It also implies extending this towards unknown people. Some argue that intergroup theory is applicable to the analysis of generalized trust due to ‘secondary transfer effects’. These occur when positive attitudes towards encountered individuals are generalized to a wider group [15,28]. If so, cross-group friendship may enhance a broader willingness to trust [29].” (p. 5)

- I have some problems disentangling trust from prejudice. In what sense these beliefs can be differentiated from the belief about someone’s trustworthiness? Some of the reasons for not being willing to send children to a school with a majority of immigrants, or to move to a district with many immigrants may be based on beliefs neither related to prejudice or to trust. For example, you may be reluctant to send your children to schools with a majority of immigrants because you have seen (in official reports, perhaps) that the quality of the education in these schools is relatively low, or you may not want to move to a district with many immigrants because you fear that this would affect to the future price of your home (perhaps not because of your prejudice but because of other people's prejudice against immigrants). Likewise, you can be in favour of giving the employer the right to only employ nationals because, as a national, it benefits you. But to the extent that these indicators do capture prejudices, in what sense do they differ from trust? It is remarkable that there is no definition of what prejudice is in the paper (neither of what trust is, for that matter). But if prejudice is the notion that someone is dishonest, or do not play by the rules of the host country, how is this different from the belief on someone’s trustworthiness? I wonder whether the results can be partly explained because what the authors are measuring is not actually prejudice, but something that combines prejudice (which, as I said, is a belief that should be very close to trusting) and other things.

These are very valid points, and we very much agree that there may well be other elements unrelated to prejudice or trust that can explain why people would be unwilling to send their kids to certain schools. Yet, the activities mentioned in the survey data we use are commonly thought to be driven in part by feelings of prejudice, and we have taken this standard operationalization to allow for comparability with previous findings. That being said, we agree that the concepts of trust and prejudice are closely related on a conceptual level, which makes it particularly important clearly to separate them in empirical work. This is the main reason why our analysis started out by testing whether our data captures two distinct concepts. We have now also provided definitions of prejudice and trust in our revised manuscript (and apologize for their absence in the previous version).

“In a first step, we assess to what extent generalized trust, threat perceptions, negative affect, and prejudice are separate latent constructs as well as how strongly these constructs are related. This is done using a Confirmatory Factor Analysis (CFA) framework, and is essential for the evaluation of the measurement part of our structural models that follow.” (p.4)

“There are important conceptual differences between generalized trust (trust towards people we do not know) [12], particularized trust (trust towards people we know) [13], and out-group trust. The latter is trust in ethnic groups one believes not to share a common descent with [2]. Generalized trust is the belief that unknown people do not do us harm [16], which differs from prejudice. Prejudice is “an antipathy based upon a faulty and inflexible generalization”[11 p. 9]. It involves the rejection of and opposition to contact with the out-group. Pettigrew and Meertens [17 p. 58] add to this definition that “prejudiced attitudes tend to form ideological clusters of beliefs that justify discrimination”.” (pp. 2-3)

Finally, it is possible that our findings reflect the operationalization of our concepts. This is by definition true for all empirical research into social and psychological concepts. Still, we should note that we follow standard operationalizations for both prejudice and trust. This implies that we are capturing the same ‘prejudice’ (whether or not combined with other things) as previous scholarship. Nonetheless, these operationalizations may be imperfect, and following your suggestion we have been more explicit about this in the revised version of the manuscript.

“Finally, it is possible that our (weak) findings are affected by the operationalization of our key concepts. While a common concerns for all empirical research into social and psychological concepts, we should note that we follow standard operationalizations for both prejudice and trust. Even though these operationalizations may be imperfect, this implies that we are capturing the same ‘prejudice’ and ‘trust’ (whether or not combined with other things) as in previous scholarship. Still, robustness check using alternative operationalizations would be important to substantiate our findings.” (p. 22)

- The authors dismiss the problem of endogeneity in a rather cavalier manner; it may be true that if the results refute the hypothesis, there is no point in conducting research with a stronger analysis; but they have to be conscious that with cross-sectional data and not causality test (such as an instrumental variable approach, for example), the results just say that it could exist a causal relationship going from cross-group friendship to prejudice, but it could also be the other way round: less prejudiced people are more likely to have more friends from other ethnic groups.

We did not mean to dismiss the problem of endogeneity, and agree that the relation between friendship and trust/prejudice may also run in the opposite direction. You are right that our data and research design cannot conclusively rule out the presence of important effects of cross-group friendship on trust. We have therefore been more careful in our wording about the implications for (not) conducting further research in this direction, as well as about the issue of causality more generally. We have added two citations that take an experimental approach for illustration (Heining Finseraas and Kotsadam 2017; Henning Finseraas et al. 2019). 

“That being said, our cross-sectional analysis cannot establish causality. Although one might pose that investigating the relations under analysis in a cross-sectional survey is futile, we believe our analysis nonetheless retains significant merit by assessing and highlighting the potential role of mediators in the contact-trust relation at the individual level.” (p.8)

“Despite these contributions, our study has also some limitations. One limitation is that the direction of the causal chain cannot be established with certainty in our dataset. While this causality issue can be overcome via panel data, the small effect sizes observed in our analysis suggest that we may need frequent measurements over a lengthy period to decompose time varying (macro and micro), and time invariant (i.e. individual-level fixed) effects. Such a panel will be a costly endeavour with potentially unsubstantial effect sizes suggested by our results. As an alternative approach, one could consider experimental research designs, which allow for much stronger causal inferences. A few recent studies have taken this route to study the relation between personal contact and anti‐immigrant sentiments as well as trust [65,66].” (p. 21).

- A minor point, the authors claim that cross-group friendship meets the condition of equal social status: Is this the case when people have a different ethnic background? Has a member of the minority group the same social status than the member of the majority?

Status is unlikely ever to be fully equal even in cross-group friendships. Yet, friendship implies that status differentials between the involved individuals are likely to be small enough (or, at least, perceived to be small enough), certainly in comparison to mere contact between individuals (where status differences may remain – and be perceived as – much more pronounced). Following your comment, we have rephrased ourselves and no longer state that cross-group friendship meets the condition of equal social status (which was too strong). Instead, we argue that friendship mitigates (perceived) status differentials and thereby goes towards meeting the condition of equal (or, at least, similar) perceived social status.

“Yet, cross-group friendship meet key requirements specified by Allport [11] as crucial conditions for the contact hypothesis to work: i.e. friendship mitigates (perceived) social status differentials between the involved individuals (…).” (p. 2)

REFERENCES

Cheung, Mike W.L. 2007. “Comparison of Approaches to Constructing Confidence Intervals for Mediating Effects Using Structural Equation Models.” Structural Equation Modeling: A Multidisciplinary Journal 14(2): 227–46.

Finseraas, Heining, and Andreas Kotsadam. 2017. “Does Personal Contact with Ethnic Minorities Affect Anti-Immigrant Sentiments? Evidence from a Field Experiment.” European Journal of Political Research 56(3): 703–22.

Finseraas, Henning et al. 2019. “Trust, Ethnic Diversity, and Personal Contact: A Field Experiment.” Journal of Public Economics 173: 72–84.

Hancock, Gregory R. 2001. “Effect Size, Power, and Sample Size Determination for Structured Means Modeling and MIMIC Approaches to between-Groups Hypothesis Testing of Means on a Single Latent Construct.” Psychometrika 66(3): 373–88.

Lakens, Daniël, Anne M. Scheel, and Peder M. Isager. 2018. “Equivalence Testing for Psychological Research: A Tutorial.” Advances in Methods and Practices in Psychological Science 1(2): 259–69.

MacKinnon, David. 2008. Introduction to Statistical Mediation Analysis Introduction to Statistical Mediation Analysis. New York, NY.: Lawrence Erlbaum Associates.

Preacher, Kristopher J., and Ken Kelley. 2011. “Effect Size Measures for Mediation Models: Quantitative Strategies for Communicating Indirect Effects.” Psychological Methods 16(2): 93–115.

Ryu, Ehri, and Jeewon Cheong. 2017. “Comparing Indirect Effects in Different Groups in Single-Group and Multi-Group Structural Equation Models.” Frontiers in Psychology 8: 747.

Sobel, Micheal E. 1987. “Direct and Indirect Effects in Linear Structural Equation Models.” Sociological Methods & Research 16(1): 155–76.

---

## [Decision Letter · Decision Letter 1]

24 Nov 2020

PONE-D-20-11120R1

Comparing the effect of cross-group friendship on generalized trust to its effect on prejudice: The mediating role of threat perceptions and negative affect

PLOS ONE

Dear Dr. Achbari,

Thank you for submitting your manuscript to PLOS ONE. After careful consideration, we feel that it has merit but does not fully meet PLOS ONE’s publication criteria as it currently stands. Therefore, we invite you to submit a revised version of the manuscript that addresses the points raised during the review process. Revisions are minor, but I agree with the few points raised by reviewer 1 and the comment of reviewer 2. Please, address these remaining questions before we proceed with the publication.

We look forward to receiving your revised manuscript.

Kind regards,

Luis M. Miller, Ph.D.

Academic Editor

PLOS ONE

Reviewers' comments:

Reviewer's Responses to Questions

**Comments to the Author**

1. If the authors have adequately addressed your comments raised in a previous round of review and you feel that this manuscript is now acceptable for publication, you may indicate that here to bypass the “Comments to the Author” section, enter your conflict of interest statement in the “Confidential to Editor” section, and submit your "Accept" recommendation.

Reviewer #1: (No Response)

Reviewer #2: All comments have been addressed

2. Is the manuscript technically sound, and do the data support the conclusions?

Reviewer #1: Yes

Reviewer #2: Yes

3. Has the statistical analysis been performed appropriately and rigorously? 

Reviewer #1: Yes

Reviewer #2: Yes

4. Have the authors made all data underlying the findings in their manuscript fully available?

Reviewer #1: Yes

Reviewer #2: Yes

5. Is the manuscript presented in an intelligible fashion and written in standard English?

Reviewer #1: Yes

Reviewer #2: Yes

6. Review Comments to the Author

Reviewer #1: I want to thank the authors for the precise answers to the previous comments and questions. They have made this second review much easier for the reviewers. That said, I think the authors did a very good job including the previous comments in the new version. I also believe the readability of the manuscript has improved a lot. I am also happy with the use of bootstrapped CIs to illustrate the difference between the coefficients. I am also quite happy with the discussion about causality and the direction of causality.

I think the paper makes a novel contribution by studying the effect of cross-group friendship in generalized trust. Although I was not convinced at the beginning that comparing this effect to the broadly studied effect of cross-friendship on prejudice was the best strategy, I think present stronger arguments in this new version of the manuscript. However, it would be nice if the authors could specify more clearly in the text that their main contribution is this new path between cross-group friendship and generalized trust. For example, the authors say in the letter:

“That being said, an important insight can be obtained from observing in our analysis that cross-group friendship is associated with lower levels of prejudice in a substantial and statistically significant manner (as also shown in previous work), while it only has a small effect on generalized trust (a new finding from our analysis).”

I believe their main contribution cannot be something that has been proved by previous literature (the link between cross-group friendship and prejudice). The main contribution is the newly discover path, which happens to be weaker than other paths as the cross-group friendship-prejudice. I would like to see this clearly stated out in the text.

The authors already talk about why generalized trust could derive from cross-group friendships (via secondary transfers). Could you please explain how the cited previous literature adresses this issue? How strong is evidence in this case?

In the new manuscript the authors mention the problem with the operationalization of the concepts and how the specific operationalization might be influencing the results. While it is important to address this concern, I think it is important to say how the specific operationalization they make can be a problem. For example, they define prejudice in the text as “an antipathy based upon a faulty and inflexible generalization”. However, the survey items used to construct prejudice do not necessarily go in this direction. For example, someone that doesn’t want to send their kids to an school in a given neighbourhood may think that schools in this area lack resources and not necessarily be prejudiced towards others. I believe this should be discuss as a limitation.

Reviewer #2: I think that you have satisfactorily answered all my queries. I still have some misgivings about the endogeneity problem, but all in all I think that the article merits publication in Plos One

7. PLOS authors have the option to publish the peer review history of their article (what does this mean?). If published, this will include your full peer review and any attached files.

Reviewer #1: No

Reviewer #2: No

---

## [Author Response · Author response to Decision Letter 1]

8 Jan 2021

Response to Reviewers

Reviewer #1: 

I want to thank the authors for the precise answers to the previous comments and questions. They have made this second review much easier for the reviewers. That said, I think the authors did a very good job including the previous comments in the new version. I also believe the readability of the manuscript has improved a lot. I am also happy with the use of bootstrapped CIs to illustrate the difference between the coefficients. I am also quite happy with the discussion about causality and the direction of causality.

Thank you for your positive evaluation of our revised manuscript and the effort we put into the revision.

I think the paper makes a novel contribution by studying the effect of cross-group friendship in generalized trust. Although I was not convinced at the beginning that comparing this effect to the broadly studied effect of cross-friendship on prejudice was the best strategy, I think present stronger arguments in this new version of the manuscript. However, it would be nice if the authors could specify more clearly in the text that their main contribution is this new path between cross-group friendship and generalized trust. For example, the authors say in the letter:

“That being said, an important insight can be obtained from observing in our analysis that cross-group friendship is associated with lower levels of prejudice in a substantial and statistically significant manner (as also shown in previous work), while it only has a small effect on generalized trust (a new finding from our analysis).”

I believe their main contribution cannot be something that has been proved by previous literature (the link between cross-group friendship and prejudice). The main contribution is the newly discover path, which happens to be weaker than other paths as the cross-group friendship-prejudice. I would like to see this clearly stated out in the text.

You are absolutely right, and our phrasing here was unnecessarily ambiguous. We wanted to indicate that the analysis of cross-group friendship and trust lies at the heart of our analysis, and that this analysis gains validity from its comparison to previously established work (on prejudice). The referee’s formulation is much more succinct and precise, and we hope (s)he does not mind that we borrowed from this in the revised version of the manuscript. Specifically, we have adjusted our formulation in the revised version of the manuscript to state explicitly in the introduction that: 

“we examine how cross-group friendship is related to generalized trust [5–9], both directly as well as indirectly via perceptions of threat and negative affect. While our main contribution lies in estimating paths between cross-group friendship and generalized trust, we also include prejudice in the model – operationalised as respondent’s desired social distance from out-group members [21,22] – as a second dependent variable. This allows us to cross-validate our findings with existing social-psychological studies, and compare the friendship-trust paths to those between friendship and prejudice to derive more detail inferences on the observed effects sizes (see below). Our main findings indicate that the relation between of cross-group friendship and trust turns out to be weaker than the relation between cross-group friendship and prejudice established in foregoing research.” (p. 4)

The authors already talk about why generalized trust could derive from cross-group friendships (via secondary transfers). Could you please explain how the cited previous literature adresses this issue? How strong is evidence in this case?

Thank you for your questions regarding our theoretical rationale. While important, prior research implicitly assumes a relationship between cross-group friendship and generalized trust, but has not tested this specific link via secondary transfers. We have included this literature, but in order not to distract from the main argument, we do not discuss the often conflicting findings in detail. As far as we are aware, only the prominent review articles by Hewstone (2015) and Dinesen et al. (2020) explicitly mention secondary transfer effects as mechanisms. We therefore derive our hypothesis 1A from the arguments in these reviews and also cite Tropp (2008) who argues for the potential of friendship in this context. Although Hewstone (2015) summarizes evidence for prejudice and outgroup trust, he only theorizes about the impact of secondary transfer effects on generalized trust. Dinesen et al’s meta-analysis reexamines the broader role of contact for trust in a multitude of contexts (e.g., neighbourhoods), but do not offer direct evidence of secondary transfer effects in terms of estimated effect sizes. Hewstone was referenced in the relevant section, but the reference to Dinesen et al. was erroneously omitted in a previous rewrite and has now been added. We thank you for helping us notice this. We note the conflicting empirical evidence for the relation between cross-group friendship and generalized trust (and hence this secondary-transfer effect), which is why we posit the alternative hypothesis 1B. As we are aware of the fact that such a path may be weaker than the established secondary transfer effects resulting in less prejudice, we have separated the trust-friendship path from the friendship-prejudice relation in this hypothesis. 

In the new manuscript the authors mention the problem with the operationalization of the concepts and how the specific operationalization might be influencing the results. While it is important to address this concern, I think it is important to say how the specific operationalization they make can be a problem. For example, they define prejudice in the text as “an antipathy based upon a faulty and inflexible generalization”. However, the survey items used to construct prejudice do not necessarily go in this direction. For example, someone that doesn’t want to send their kids to an school in a given neighbourhood may think that schools in this area lack resources and not necessarily be prejudiced towards others. I believe this should be discuss as a limitation.

We thank the reviewer for this insightful example. We totally agree that the construct validity of this specific item is a limitation we should have discussed, and have therefore added this to our manuscript (please see below). Having said this, we still believe there is merit in keeping to standard operationalizations for comparison sake and that we have included many items in our scale, which together can tap into a wider antipathy and social distance towards immigrants as out-groups. 

“Finally, it is possible that our (weak) findings are affected by the operationalization of our key concepts. For example, our prejudice scale includes an item on (un)willingness to send children to a majority immigrant school, which could be based on reasonable concerns about the lack of resources rather than inflexible generalizations about out-groups. While a common concern for all empirical research into social and psychological concepts, we should note that we follow standard operationalizations for both prejudice and trust. Even though these operationalizations may be imperfect, this implies that we are capturing the same ‘prejudice’ and ‘trust’ (whether or not combined with other things) as in previous scholarship. Still, robustness checks using alternative operationalizations would be important to substantiate our findings.” (p. 22)

Reviewer #2: 

I think that you have satisfactorily answered all my queries. I still have some misgivings about the endogeneity problem, but all in all I think that the article merits publication in Plos One

Thank you for your assessment that our revised manuscript merits publication in PLOS One, and for your recognition of the effort we put into the revision. Naturally, we understand your continuing concern about endogeneity, which is always a key issue in empirical work on the type of data available to us. Following your comment as well as a suggestion from the editor, we have therefore further clarified our discussion on this important point in the revised manuscript. Specifically, we now introduce our modelling approach stating that:

“we test our hypotheses by evaluating direct, indirect, and total (indirect) effect in a set of Structural Equation Models. Strictly speaking, we cannot speak of “effects” since we employ cross-sectional data and the data available to us do not allow a credibly resolution to the endogeneity concerns arising in our setting. As such, the analysis below cannot confirm the direction of causality. While we use Structural Equation Modelling terminology to avoid confusion, it should be kept in mind that our analyses merely establish the presence (or absence) of correlations between the variables under analysis.” (p. 13)

We also revised our discussion of this important point in the conclusion further to stress its importance to readers. Crucially, we thereby point to a number of potential avenues to take our research further (i.e. panel data as well as experiments).

“One persistent and unaddressed concern relates to a potential endogeneity problem. Our dataset is indeed unable to establish (the direction of) causal chains with certainty. While such endogeneity and causality concerns can be overcome via panel data, the small effect sizes observed in our analysis suggest that we may need frequent measurements over a lengthy period to decompose time varying (macro and micro), and time invariant (i.e. individual-level fixed) effects. As an alternative approach, one could consider experimental research designs, which allow for much stronger causal inferences. A few recent studies have taken this route to study the relation between personal contact and anti‐immigrant sentiments as well as trust [66,67].” (p. 21)

---

## [Editor Report · Decision Letter 2]

12 Jan 2021

Comparing the effect of cross-group friendship on generalized trust to its effect on prejudice: The mediating role of threat perceptions and negative affect

PONE-D-20-11120R2

Dear Dr. Achbari,

We are pleased to inform you that your manuscript has been judged scientifically suitable for publication and will be formally accepted for publication once it meets all outstanding technical requirements.

Kind regards,

Luis M. Miller, Ph.D.

Academic Editor

PLOS ONE

---

## [Editor Report · Acceptance letter]

27 Jan 2021

PONE-D-20-11120R2 

Comparing the effect of cross-group friendship on generalized trust to its effect on prejudice: The mediating role of threat perceptions and negative affect 

Dear Dr. Achbari:

I'm pleased to inform you that your manuscript has been deemed suitable for publication in PLOS ONE. Congratulations! Your manuscript is now with our production department. 

Kind regards, 

on behalf of

Dr. Luis M. Miller 

Academic Editor

PLOS ONE